# Synthesizing Bonds: Enhancing Adult Attachment Predictions with LLM-Generated Data

## Abstract

Obtaining data in the medical field is challenging, making the adoption of AI technology within the space slow and high-risk. We evaluate whether we can overcome this obstacle with synthetic data generated by large language models (LLMs). In particular, we use GPT-4 and Claude 3 Opus to create agents that simulate adults with varying profiles, childhood memories, and attachment styles. These agents participate in simulated Adult Attachment Interviews (AAI), and we use their responses to train models for predicting their underlying attachment styles. We evaluate our models using a transcript dataset from 9 humans who underwent the same interview protocol, analyzed and labeled by mental health professionals. Our findings indicate that training the models using only synthetic data achieves performance comparable to training the models on human data. Additionally, while the raw embeddings from synthetic answers occupy a distinct space compared to those from real human responses, the introduction of unlabeled human data and a simple standardization allows for a closer alignment of these representations. This adjustment is supported by qualitative analyses and is reflected in the enhanced predictive accuracy of the standardized embeddings.

## 1 Introduction

Acquiring medical data from human subjects is challenging due to ethical and logistical concerns. Strict adherence to regulations like the Health Insurance Portability and Accountability Act (HIPAA) in the U.S. and the General Data Protection Regulation (GDPR) in Europe is crucial for safeguarding patient privacy and confidentiality, and businesses face significant penalties for mishandling sensitive data.

Collecting data can be a resource-intensive and time-consuming process that requires patient consent and commitment, continuous iterations with institutional review boards (IRBs), and researcher availability to ensure data accuracy and reliability. In addition, early capital companies may lack the resources to initiate or last until they have access to meaningful and substantial data necessary to address patient problems, which can involve complex and data-intensive processes. Lastly, studying the impact of therapeutic interventions can be remarkably delicate, especially in the context of mental health, where some treatments may have adverse effects, and successful outcomes often depend on personalized and patient-specific approaches.

Possible solutions include utilizing public data sources like the UK Biobank (Sudlow et al., 2015) or the All of Us research program (All of Us Research Program Investigators, 2019), though these are restricted by data scope and licensing constraints. Alternatively, using private datasets is costly which can limit scalability. Conducting more experiments is another option, but it is time-consuming and resource-intensive. Lastly, simulating real data with synthetic data is feasible, but creating convincing and accurate synthetic data poses significant challenges.

In this work, we explore modern advances in large language models (LLMs) to embody characteristics of real human behavior in the mental health space. The mental health sector presents a unique opportunity for LLMs, as many therapies primarily involve conversational interactions. Improvements in LLMs have enhanced our understanding of language, opening up new possibilities for addressing mental health challenges.

This approach has several benefits, including preserving patient confidentiality, reducing the need for direct patient involvement, and allowing researchers to investigate different scenarios without the ethical and privacy constraints associated with real human data. Most importantly, it can overcome the challenges of limited data, often confined within mental health institutions, allowing researchers to experiment with data-intensive machine-learning techniques that would be impractical with only human-derived data.

We focus on applying synthetic data to understand attachment styles. Attachment styles delineate how people form and maintain emotional bonds with others, primarily developed through early relationships with caregivers. They are a fundamental aspect of human psychological development, influencing behaviors and interactions throughout one's life and making them a significant indicator of potential psychiatric traits. For instance, understanding an individual's attachment style can help predict various mental health outcomes, such as susceptibility to anxiety, depression, and personality disorders (Gerber, 2005; Mikulincer & Shaver, 2012). Consequently, attachment style serves as a practical basis for both assessing risk factors and tailoring interventions in mental health care, enhancing the precision and effectiveness of psychiatric treatments.

A common psychiatric practice for assessing attachment style in adults is the Adult Attachment Interview (AAI). It is a structured interview in which individuals reflect on their childhood experiences and relationships with caregivers, allowing researchers to categorize their attachment style based on their responses. In this work, we create artificial agents powered by State-of-the-Art LLMs to simulate human participants undergoing AAIs. We then use the synthetic transcripts from these dialogues to train machine learning models and test their predictive performance on a separate set of real human transcripts from humans undergoing the same interview protocol, which mental health professionals have already assessed for attachment style. Our results indicate that training a model with synthetic transcripts can effectively predict the attachment styles of real individuals using three different classifiers. Moreover, qualitative analysis shows that with a straightforward correction, embeddings used to encode synthetic transcripts form clusters that seem to align with attachment styles in real humans.

It is important to mention that the challenges in the mental health space extend beyond predicting attachment styles. They apply to most tasks aimed at predicting aspects of human behavior. We utilize synthetic data to explore the prediction of attachment styles as an initial proof of concept, demonstrating that similar approaches can benefit from the findings here in scenarios beyond the attachment theory domain where human data is scarce or of limited access.

Our key contributions include: 1) Developing an agent architecture that simulates humans with different user profiles and childhood memories; 2) Demonstrating that training models with data from these artificial agents is predictive of attachment styles in real humans, as evidenced by fitting three different traditional classifiers with synthetic data; 3) Showing that unlabeled data can be employed to standardize the raw embeddings of synthetic data, creating a more uniform representation between synthetic and human data thereby improving prediction performance.

## 2 RELATED WORK

Previous studies have utilized generative AI and LLMs in healthcare to enhance data management, information retrieval, decision-making processes, and synthetic data generation (Yu et al., 2023; Nassiri & Akhloufi, 2024; Magister et al., 2023). For example, the research by Tang et al. (2023) employs few-shot learning to create labeled datasets for biological named entity recognition and relation extraction. This method addresses privacy concerns by circumventing the need to upload sensitive patient data for text mining directly.

LLMs have also been applied in synthetic data generation for tasks such as mathematical reasoning (Saxton et al., 2019; Lozhkov et al., 2024), and image captioning (Hammoud et al., 2024), among others. In the mental health space, much of the focus has been on developing chatbots that facilitate health communication, with evaluations primarily concerning the ability of LLMs to deliver high-quality, empathetic responses to patient queries (Ayers et al., 2023; Ma et al., 2024; Cabrera et al., 2023; Cai et al., 2023; Fu et al., 2023). However, there is a growing body of research that employs LLMs for synthetic data generation to explore various aspects of human behavior (Jiang et al., 2024;

Serapio-García et al., 2023; Jiang et al., 2023) and mental conditions such as depression (Mori et al., 2024; Chen et al., 2023; Lan et al., 2024).

To our knowledge, there has been no use of LLMs to study adult attachment style in real humans, with prior related efforts focusing on analyzing attachment style using real data sources such as EEG signals (Laufer et al., 2024), audio waves (Koçak et al., 2023; Gomez-Zaragoza et al., 2023), and text transcripts (Calvo, 2018).

Inspired by prior work on LLM-empowered synthetic agents with varied roles and traits (Park et al., 2023; Qian et al., 2023), we have developed synthetic agents with diverse profiles and childhood memories. We use these agents to generate synthetic transcripts, which we then use to train models that predict attachment styles in real humans, thereby extending the use of LLMs into the domain of psychological research and attachment theory.

## 3 ATTACHMENT STYLE

Attachment theory (Ainsworth & Bowlby, 1991) provides a framework to understand the dynamics of interpersonal relationships, focusing on the emotional bonds between individuals. Central to this theory is the concept of attachment styles, which are patterns of relating to others. These patterns are formed early in life and extended into adulthood. A typical categorization comprises four primary types: *avoidant*, *secure*, *anxious*, and *fearful-avoidant*. Here, we focus on an alternative traditional categorization comprised of three classes: *avoidant*, *secure*, and *preoccupied* behavior (Ainsworth et al., 1978).

Secure attachment is characterized by a positive view of oneself and others, leading to healthy, trusting relationships. Avoidant attachment is marked by discomfort with closeness and a tendency to maintain emotional distance. Preoccupied attachment involves hypersensitivity to separation and an intense fear of abandonment (Ainsworth et al., 1978).

An Adult Attachment Interview (AAI) is a standard tool to assess an individual's attachment style in adulthood. The interviews follow a structured protocol that encourages people to reflect on their childhood experiences and perceptions of parental relationships (George et al., 1985). Psychologists can classify a person's attachment style through careful analysis of the responses, particularly the coherence and emotional content. The AAI has been instrumental in linking adult attachment styles with early parental interactions and predicting behavior patterns in adult relationships (Hazan & Shaver, 2017).

Recognizing someone's attachment style can be advantageous in the context of mental health treatment, for example, as it can help therapists understand the root of relational difficulties and emotional disturbances that may arise in therapy. Knowing that a patient has an avoidant attachment style can alert a therapist to the potential challenges in establishing a close therapeutic relationship. It can also help with the development of therapies that are more tailored to the individual's needs.

However, predicting someone's attachment style presents several challenges, primarily because attachment styles are complex and influenced by many factors. In particular, attachment styles can change over time due to new relationships, life experiences, or personal growth. Additionally, people may not always have accurate self-awareness or may present themselves in a biased way, unintentionally or intentionally. Lastly, attachment behaviors can sometimes be similar to or influenced by other psychological conditions like anxiety, depression, or personality disorders. This overlap can make it difficult to discern whether attachment issues or other psychological challenges drive specific behaviors.

## 4 SYSTEM ARCHITECTURE

To generate synthetic data, we developed artificial agents equipped with a basic brain structure, memory, and language capabilities, as shown in Figure 1.

The process begins with the creation of unique profiles for each artificial agent. These profiles include details such as name, age, gender, race, current occupation, place of birth, date of birth, past residences, and three descriptive adjectives for each parent and their occupations. We use GPT-4 with

**Interviewee Agent Creation**          **Simulated AAI**

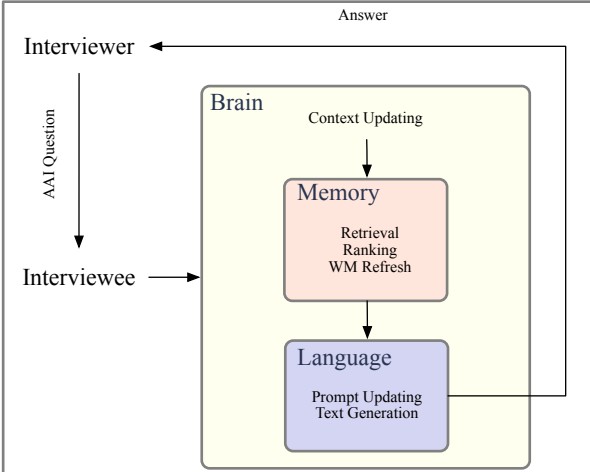

Figure 1: The main components of the system. It begins with the creation and persistence of interviewee agents, each equipped with a unique profile and ten childhood memories. During a simulated Adult Attachment Interview (AAI), an interviewee agent updates its working memory (WM) based on the current context, which includes previous chat messages. The agent retrieves and ranks relevant childhood memories, which are then fed into the language module. This module incorporates the selected memories, the user profile, chat history, and the most recent AAI question into the prompt to generate an appropriate response. This process is repeated for each question in the interview until there are no more questions to ask.

a temperature setting of 0.7 to generate these profiles, using the instructions specified in Appendix A. Figure 2 displays a sample profile produced with this method.

```
{
    "name": "Leonard Fitzgerald",
    "age": 45,
    "race": "Caucasian",
    "gender": "Male",
    "dob": "1976-03-14",
    "birthplace": "Place 567",
    "current_job": "Architect",
    "places_lived": "Place 567, Place 233, Place 899",
    "children": "2 sons, 1 daughter",
    "siblings": "1 brother, 2 sisters",
    "fathers_jobs": "Construction worker, Carpenter",
    "mothers_jobs": "Nurse, School teacher",
    "father_adjectives": "Hardworking, Strict, Impatient",
    "mother_adjectives": "Compassionate, Understanding, Overprotective"
}
```

Figure 2: Example of a user profile generated by GPT-4 with instructions in Appendix A.

Following the generation of user profiles, we integrate them into the instructions described in Appendix B and prompt an LLM (we used chat GPT-4) to produce a collection of 10 childhood memories. We adjust the temperature to 0.7 to generate more creative memories.

These childhood memories give the agents a richer context, enabling them to deliver more precise interview responses and avoid generic or incoherent replies akin to answers that might be expected from actual humans. Also, we aim to enhance the diversity of the synthetic interviews, as each agent

will possess a distinct profile and set of life experiences. Figure 3 presents an example of a childhood memory generated for the profile depicted in Figure 2.

```
{
    "creation_timestamp": "1985-05-12 20:00:00",
    "content": "My father scolded me harshly for not finishing my
    homework on time. He was strict and impatient, and his outburst
    made me feel small and rejected. I remember retreating to my room,
    vowing to myself that I would prove him wrong."
}
```

Figure 3: Example of a childhood memory generated by GPT-4 for the user profile in Figure 2 with instructions in Appendix B.

We store the childhood memories generated in a vector store database. During interviews, we use Retrieval-Augmented Generation (RAG) to fetch three of these memories based on the context of the four most recent chat messages. We use the HuggingFace all-MiniLM-L6-v2 model to embed these memories and employ cosine similarity for retrieval. We use RAG here instead of copying all the memories to keep the prompt as short as possible and avoid feeding information irrelevant to answering the question. The choice of embedding model was arbitrary. One could employ better private embedding models. However, we observed that with the HuggingFace model, we could retrieve memories compatible with the questions asked (e.g., retrieval of memories about the parents when the question involved the father or mother).

We created 60 artificial agents with unique profiles and 10 associated childhood memories each. These agents engage in simulated interviews with an interviewer agent, taking turns to chat. The interviewer follows the Adult Attachment Interview (AAI) protocol, utilizing a predefined list of questions, which ensures a structured and consistent inquiry. We present the complete list of questions in Appendix C.

During an interview, each message received by an interviewee agent is passed to its "brain". This component then updates its context with the interviewer's question and prompts the memory module to refresh the agent's working memory based on this context. This involves retrieving relevant childhood memories from the vector store, using the current chat context as a query, and ranking based on the associated retrieval score.

These selected memories are then loaded into the agent's working memory and included in the prompt used to generate the next response. This prompt also contains the agent's profile and specific instructions to reflect an attachment style, detailed in Appendix D.

We use these agents to produce two synthetic datasets containing AAI transcripts generated by agents powered by GPT-4 and Claude 3 Opus models. During the dialogues, we set both models to a temperature of 0.5 on the intuition that we do not want the responses to be the same (temperature 0), and we want to avoid responses that are too far from the context of the interview question due to potential hallucination (high temperature).

## 5    EVALUATION

We evaluate synthetic data on a task that involves predicting attachment styles from transcripts of real human responses to Adult Attachment Interviews (AAI). For this, we train models on synthetic transcripts created by language model-empowered agents responding to identical AAIs.

### 5.1    DATASETS

We produced two synthetic datasets: one using a GPT-4 model and the other employing a Claude 3 Opus model. Each dataset consists of 60 interviews, with 20 representing each attachment style, answered by distinct artificial agents as detailed in section 4. Every interview comprises a fixed sequence of 19 questions and responses.

Table 1: Number of interviews per attachment style in the human and synthetic datasets.

| Attachment Style | Labeled Human | Unlabeled Human | Synthetic |
|---|---|---|---|
| Avoidant | 4 | n.a. | 20 |
| Preoccupied | 3 | n.a. | 20 |
| Secure | 2 | n.a. | 20 |
| Total | 9 | 17 | 60 |

The dataset derived from humans consists of transcripts from the Adult Attachment Interview (AAI) by Main & Goldwyn (1994), utilized in a more extensive study on psychotherapeutic processes and outcomes involving 26 young adults at the Anna Freud Centre in London, England (Gerber, 2005). During that study, health professionals following standard procedures for classifying attachment styles labeled nine of those interviews, which we used to assess the performance of our predictive models. The remaining 17 transcripts are unlabeled human data we used for a standardization technique discussed in section 6. Table 1 displays the breakdown of interviews by attachment style across all datasets.

To prepare the data for analysis, we cleaned the original human-derived transcripts by removing linguistic markers such as ellipses ("...") and dashes ("–"), which the transcriber used to indicate pauses. We also excluded responses that contained fewer than ten words. Unlike the synthetic dataset, the human dataset interviews do not have a predetermined number of questions; interviewers could ask additional questions to delve deeper into specific topics or clarify responses. We keep these supplementary questions in the dataset.

For both the synthetic and human-derived datasets (including both labeled and unlabeled data), we employed OpenAI's text-embedding-3-small model to create a 1536-dimensional embedding vector for each transcribed answer. We then calculated the interview embedding vector by taking the element-wise average of all answer embedding vectors within a single interview.

We encode the responses with the OpenAI embedding model instead of the HuggingFace model we used to embed memories because the agent responses can be arbitrarily longer than the memories. The OpenAI model has better benchmark performance than the HuggingFace model (Muennighoff et al., 2023), suggesting the former can capture the semantics of the answers better than the latter.

## 5.2 RESOURCES

We generate synthetic data by making API calls to private LLMs. Predictive analysis on the generated datasets uses simple classifiers and consumes very few resources. During synthetic data generation, we consumed about 1.6M input tokens, 200K output tokens with the GPT-4 model, and 500K output tokens with the Claude 3 Opus model. Each interview took about 2 minutes to be completed.

## 5.3 CLASSIFIERS

Our experiments use the scikit-learn 1.4.2 implementation of logistic regression, extra trees, and multilayer perceptron (MLP) classifiers. We normalize the features to have mean zero and unitary standard deviation before model fitting. For linear regression, we use *liblinear* solver and evaluate the performance using $\ell 1$ and $\ell 2$ penalties to mitigate potential overfitting, especially in experiments using only the human dataset, which is quite small. We set the number of estimators in the Extra Trees model to 500 and the number of iterations in the MLP model to 1000. We leave the remaining parameters at their defaults.

## 6 EXPERIMENTS

We divide our experiments into two phases to establish the integrity of our synthetic data before leveraging it to enhance predictive models. Initially, we conduct a qualitative assessment of the diversity and quality of our synthetic datasets. Following this, we analyze the extent to which synthetic data can help predict attachment styles in real humans.

## 6.1 SYNTHETIC DATA ANALYSIS

As previously mentioned, we implemented mechanisms like creating different user profiles and childhood memories to increase diversity in the synthetic dataset. In our first experiment, we assess how diverse the synthetic data is, which is essential to establish that any improvement in downstream tasks due to adding more synthetic interviews reflects genuine information gained in the model.

We check diversity by computing cosine similarities between vector embeddings from all pairwise combinations of interviews in the same attachment style. Figure 4 shows the distributions of these similarities for interviews generated by GPT-4 and Claude 3 Opus. Similarities are high, suggesting that interview embeddings from the same attachment style are clustered together. However, they are not the same, as indicated by the variance in the distributions, which shows that the different synthetic interviews have some diversity in the embedding space.

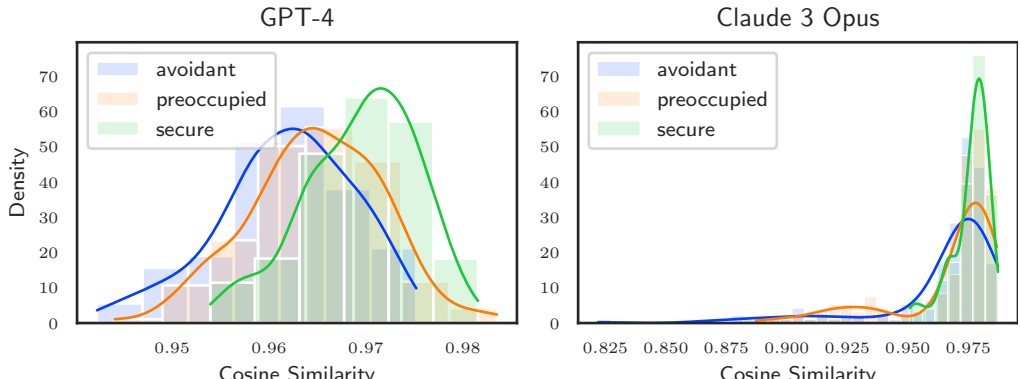

Figure 4: The distribution of cosine similarities between all pairwise combinations of embeddings in the synthetic datasets generated by two different large language models (LLMs). We computed cosine similarities per attachment style.

To evaluate the similarity between synthetic and human data in the embedding space, we project the original 1536-dimensional embeddings from both artificial and human-derived datasets (labeled and unlabeled) into a 2D space using Uniform Manifold Approximation and Projection (UMAP) (McInnes et al., 2018). This approach reveals distinct clusters within the synthetic dataset's embeddings for each attachment style as illustrated in Figure 5. However, these clusters occupy different regions of the space compared to the human-derived data embeddings.

To address this, we implement a standardization technique to align the synthetic embeddings more closely with the human data. The procedure involves calculating a vector $u$ as the element-wise mean of all embeddings in the unlabeled human dataset and a similar mean vector $s$ for the synthetic embeddings. We then adjust each synthetic embedding by adding $u - s$ to it. We apply this procedure to the original embeddings before projection using UMAP.

This standardization method effectively shifts the synthetic embeddings nearer to those of the human data as depicted in Figure 5 for the GPT-4 synthetic dataset. Moreover, the attachment style clusters within the normalized synthetic data show alignment with the attachment styles in the labeled human data despite this dataset not being used in either the synthetic data generation or the embedding normalization process. For comparison, see Figure 10 in Appendix E, which illustrates a similar pattern with synthetic interviews generated by the Claude 3 Opus model.

## 6.2 ATTACHMENT STYLE PREDICTION

Next, we evaluate our ability to predict attachment styles using only synthetic data. We compare it against a baseline model trained with leave-one-out cross-validation (CVLOO) on the human-labeled dataset. For models trained on synthetic data, we include results for both standardized and non-standardized synthetic embeddings. The results are in Table 2.

Models trained on standardized synthetic data surpassed those trained with a limited number of human-derived interviews in almost all cases. However, CVLOO in the human dataset is prone to

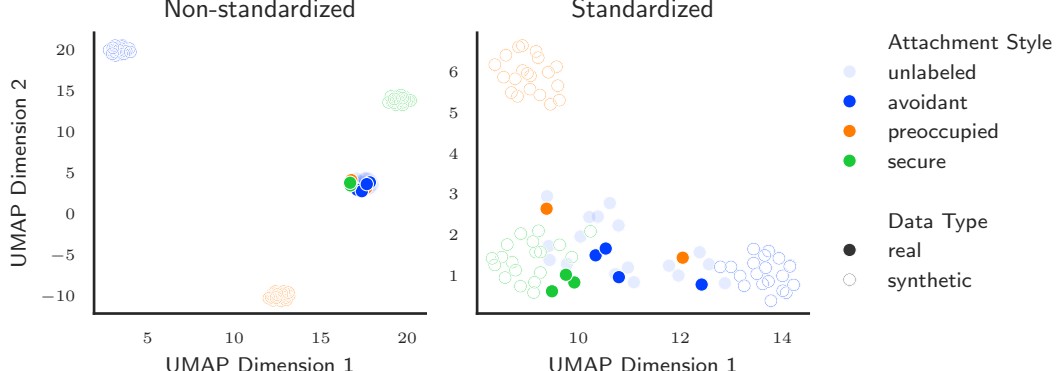

Figure 5: 2D UMAP projections of synthetic (GPT-4) and human data embeddings. Both plots show clear clusters corresponding to the different attachment styles within the synthetic embeddings. In the left plot, however, the synthetic embeddings occupy a distinct region of the space, separate from the human embeddings. The right plot demonstrates the impact of standardizing synthetic embeddings using unlabeled human data, where the synthetic embeddings are now more closely grouped, and the attachment style clusters better align with the attachment styles of the labeled human dataset interviews.

overfitting, given the small size of this dataset and the high dimensionality of the input embeddings. Results using LR with $\ell 1$ penalty, which tends to create more sparse features, suggest that this is the case, as we see performance comparable to the best synthetic data figures when we use this penalty. Additionally, standardizing the synthetic data with unlabeled human data substantially enhances performance, corroborating our qualitative observations in Figure 5 and Figure 10.

Table 2: ROC AUCs and standard errors (in parentheses) for predicting attachment styles in the human dataset across different classifiers using standardized and non-standardized 1536-dimensional representations of each interview as input instances. We compute the ROC AUC in the Human dataset with leave-one-out (CVLOO) by concatenating the predictions of each test split (held-out interview) and computing the ROC AUC once using all nine outcomes. We estimate standard errors in the extra trees and MLP models by training and testing these models with ten different random seeds.

| Training Source | LR ($\ell 1$) | LR ($\ell 2$) | Extra Trees | MLP |
|---|---|---|---|---|
| Human (LOO) | 0.74 | 0.68 | 0.62 (0.02) | 0.68 (0.04) |
| GPT-4 | 0.64 | 0.67 | 0.62 (0.01) | 0.62 (0.02) |
| GPT-4 (standardized) | 0.69 | 0.75 | 0.76 (0.01) | 0.71 (0.04) |
| Claude 3 Opus | 0.64 | 0.66 | 0.69 (0.01) | 0.67 (0.02) |
| Claude 3 Opus (standardized) | 0.73 | 0.78 | 0.72 (0.01) | 0.74 (0.03) |

Next, we evaluate how the volume of synthetic data influences the results. For this analysis, we use standardized synthetic embeddings to train the models on synthetic data, since they have shown the best performance in our previous experiment.

Our procedure consists of sampling $n$ synthetic interviews per attachment style, standardizing their embeddings, and fitting various models. We repeat this sampling process ten times and calculate the mean ROC AUC and standard error to reduce potential bias in the results due to selecting specific interviews. We then gradually increase $n$ and plot the ROC AUC curves to determine if additional synthetic data enhances model performance. We present the resulting plots in Figure 6.

We do not observe an increasing trend in performance as we train the models with more synthetic samples besides the initial increment. We believe this is the case because the synthetic data is fairly easy to classify with just a few samples, which is evidenced by the well-defined clusters we observe in Figure 5 and Figure 10. Fluctuating moods and mixed emotions can influence human interview responses, which may obscure the attachment styles reflected in human answers. In contrast, driven

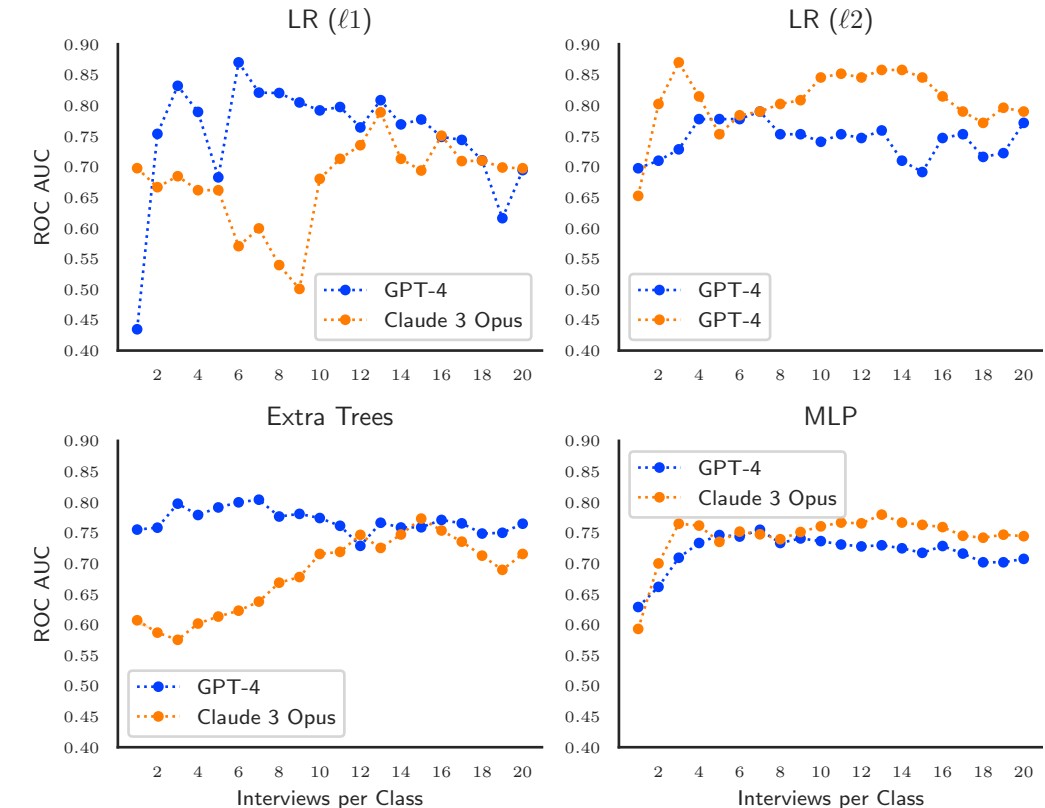

Figure 6: ROC AUCs for predicting attachment styles in the human dataset by training different models on an increasing number of synthetic interviews generated by two distinct large language models (LLMs). We trained the models using synthetic embeddings standardized with unlabeled human data. The bands displayed around each point on the plot represent the standard error of the ROC AUCs, calculated from ten different samplings of the interviews associated with the results in each data point.

by instructions, synthetic agents more consistently embed their underlying attachment styles into their responses, facilitating a more pronounced delineation of the decision boundary between the different attachment styles.

## 7   LIMITATIONS

We found that synthetic data could predict human attachment styles effectively. However, performance is limited by how easy it is to classify synthetic samples, so alternative ways to create more ambiguous responses that better resemble human behavior are warranted. Also, the scarcity of labeled human interviews may have led to underestimations in predictions from models trained on human data. We did not formally evaluate how closely synthetic responses mimic human ones, nor did we explore using open-source models or fine-tuning techniques, which might yield more realistic responses. Additionally, we did not thoroughly investigate other aspects of human behavior to see how broadly this approach could be applied.

## 8   SOCIETAL IMPACTS

This research has implications for mental health and AI ethics. We demonstrated that we could accurately predict human attachment styles from machine learning models trained on synthetic conversational data generated by large language models (LLMs). This finding could help clinicians better understand patients' interpersonal issues, enabling personalized treatment. However, employing

synthetic data and AI for psychological profiling poses ethical risks, such as potential misuse for unintended purposes. Additionally, while synthetic data reduces privacy concerns, ensuring that these models do not reinforce biases or inaccuracies is essential, especially when real human data is scarce or biased.

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

SUPPLEMENTARY MATERIAL

## A    USER PROFILE PROMPT

We employ the prompt in Figure 7 to create a random user profile. We carefully chose the components of this profile to align with the questions the agent must address during a simulated Adult Attachment Interview (AAI) session.

```
Generate text in JSON format containing random detailed personal and
familial information for a fictional individual. Be creative. The JSON
should include the following keys and their corresponding values:

1. `name`: random first and last names for the person.
2. `age`: The person's age.
3. `race`: The person's race.
4. `gender`: The person's gender (either male, female, gay or
             non-binary).
5. `dob`: The person's birthdate in the format YYYY-MM-DD.
6. `birthplace`: The location of the person's birth in the format
                 Place X where X is a random number.
7. `current_job`: The person's current occupation.
8. `places_lived`: A comma-separated list of places where the person
                   has lived in the format Place X, Place Y where X
                   and Y are random numbers.
9. `children`: A comma-separated list of children the person has
               (e.g., 1 son and 1 daughter). Do not include names.
10. `siblings`: A comma-separated list of siblings the person has
                (e.g., 1 sister). Do not include names.
11. `fathers_jobs`: A comma-separated list of jobs held by the
                    person's father.
12. `mothers_jobs`: A comma-separated list of jobs held by the
                    person's mother.
13. `father_adjectives`:  A comma-separated list of three adjectives
                          describing the person's father. Include
                          negative adjectives.
14. `mother_adjectives`: A comma-separated list of three adjectives
                         describing the person's mother. Include
                         negative adjectives.

Each field should be filled with realistic, coherent data that aligns
with the person's background and life story. Ensure the information
provided is consistent and plausible, reflecting a believable
character profile.
```

Figure 7: Prompt used to generate a random user profile.

## B CHILDHOOD MEMORIES PROMPT

We generate childhood memories using the prompt in Figure 8. This prompt includes several placeholders filled at the time of generation. Specifically, we substitute *reference_timestamp* with the timestamp at the time of generation and *user_profile* with a JSON object containing a user profile. We explicitly instruct the LLM to cover topics addressed by the questions the agent must respond to during a simulated Adult Attachment Interview (AAI) chat.

```
#INSTRUCTIONS#
You are a human with the #PROFILE# below.
Generate 10 #CHILDHOOD MEMORIES# as a JSON object containing a list of
JSON objects.
Each memory must contain rich, vivid and concrete situations you lived
with your parents or siblings.
Each memory must always depict event as in the #EXAMPLES#.
Do not reinstate your parents' jobs or adjectives on the memories.
Your memories must not be overly optimistic.
Your memories should cover all the #TOPICS# below.
Your memories are written as a reenactment of the scene as in
#EXAMPLES#.
Your memories must include traumatic events like abuse, death or
severe diseases like cancer.
Your memories about your parents must be consistent with their jobs
and description.
Each memory is a JSON object with the following fields:

1. `creation_timestamp`: When the event happened in the format
YYYY-MM-DD HH:mm:SS given that today is {reference_timestamp}.
2. `content`: The content of the memory.

#EXAMPLES#
- My father got home and spanked me. He was yelling as soon as he got
home for no reason. He came for me and my mom was helpless. I was just
a child, she should have helped me. This was not the only time he did
this, but that day he left physical scars in my body.
- I got home from school and my mom was missing. She didn't tell us
where she went. I got really scared and anxious she left us for good.
- The police called to tell us my brother died in a car crash but my
dad didn't drop a single tear. My mom cried like a baby and I felt
like my world was ending.

#PROFILE#
{user_profile}

#TOPICS#
Family background and early childhood living situation
Specific incidents with parents based on their personality
Coping mechanisms during childhood distress
Reactions to physical injury during childhood
Feelings of rejection during childhood
Relationships with other significant adults during childhood
Experience of losing a parent or close loved one in childhood and
adulthood
Traumatic experiences

#CHILDHOOD MEMORIES#
```

Figure 8: Prompt used to generate 10 random childhood memories for a user profile.

## C  AAI QUESTIONS

1. Could you start by helping me get oriented to your early family situation, and where you lived and so on? If you could tell me where you were born, whether you moved around much, what your family did at various times for a living?

2. I'd like you to try to describe your relationship with your parents as a young child if you could start from as far back as you can remember?

3. Now I'd like to ask you to choose five adjectives or words that reflect your relationship with your mother starting from as far back as you can remember in early childhood–as early as you can go, but say, age 5 to 12 is fine. I know this may take a bit of time, so go ahead and think for a minute...then I'd like to ask you why you chose them. I'll write each one down as you give them to me.

4. Now I'd like to ask you to choose five adjectives or words that reflect your childhood relationship with your father, again starting from as far back as you can remember in early childhood–as early as you can go, but again say, age 5 to 12 is fine. I know this may take a bit of time, so go ahead and think again for a minute...then I'd like to ask you why you chose them. I'll write each one down as you give them to me.

5. Now I wonder if you could tell me, to which parent did you feel the closest, and why? Why isn't there this feeling with the other parent?

6. When you were upset as a child, what would you do?

7. Did you ever feel rejected as a young child? Of course, looking back on it now, you may realize it wasn't really rejection, but what I'm trying to ask about here is whether you remember ever having rejected in childhood

8. Were your parents ever threatening with you in any way - maybe for discipline, or even jokingly?

9. In general, how do you think your overall experiences with your parents have affected your adult personality?

10. Why do you think your parents behaved as they did during your childhood?

11. Were there any other adults with whom you were close, like parents, as a child?

12. Did you experience the loss of a parent or other close loved one while you were a young child–for example, a sibling, or a close family member?

13. Other than any difficult experiences you've already described, have you had any other experiences which you should regard as potentially traumatic?

14. Now I'd like to ask you a few more questions about your relationship with your pants. Were there many changes in your relationship with your parents (or remaining parent) after childhood? We'll get to the present in a moment, but right now 1 mean changes occurring roughly between your childhood and your adulthood?

15. Now I'd like to ask you, what is your relationship with your parents (or remaining parent) like for you now as an adult? Here I am asking about your current relationship.

16. I'd like to move now to a different sort of question–it's not about your relationship with your parents, instead it's about an aspect of your current relationship with other relatives. How do you respond now, in terms of feelings, when you separate from your child / children?

17. If you had three wishes for your child twenty years from now, what would they be? I'm thinking partly of the kind of future you would like to see for your child I'll give you a minute or two to think about this one.

18. Is there any particular thing which you feel you learned above all from your own childhood experiences? I'm thinking here of something you feel you might have gained from the kind of childhood you had.

19. We've been focusing a lot on the past in this interview, but I'd like to end up looking quite a ways into the future. We've just talked about what you think you may have learned from your own childhood experiences. I'd like to end by asking you what would you hope your child (or, your imagined child) might have learned from their experiences of being parented by you?

# D   CHAT PROMPT

The prompt an agent uses to respond to a simulated Adult Attachment Interview (AAI) question contains placeholders within curly brackets, which the agent's language module substitutes with specific values for different agents. The *attachment_style_description* placeholder changes based on the agent's attachment style and can be one of the following options:

**Avoidant** "You tend to maintain emotional distance, minimize closeness, and often reject or withdraw from intimacy in relationships. You have a selective memory, often downplaying or dismissing past experiences involving intimacy or vulnerability".

**Preoccupied** "You have a heightened need for reassurance, fear of abandonment, and a constant seeking of closeness and validation in relationships. You dwell on past experiences, focusing on moments of insecurity or inconsistency in relationships".

**Secure** "You are comfortable with intimacy, have balanced independence, effective communication, and a sense of safety and trust in relationships. You view your memories through a lens of safety and trust".

In the profile section of the prompt, the agent's language module fills placeholders based on information from a JSON object that describes the user's profile. It substitutes *life_memories* with memories relevant to the question asked, retrieved using a Retrieval-Augmented Generation (RAG) technique during the interview. The *chat_history* placeholder contains the previous four chat messages to maintain context.

```
You are an assistant impersonating a human with a given #PROFILE#
chatting with another human.
You can use the events in #MEMORY# to talk about concrete experiences
you lived.
Respond in a way that matches this description without explicitly
saying the way you feel "{attachment_style_description}". You will be
penalized if you mention any part of this description in your answer.
Your answer must be 1-paragraph long and you must not ask questions.

#PROFILE#
name: {name}
age: {age}
dob: {dob}
current occupation: {current_job}
birthplace: {birthplace}
children: {children}
siblings: {siblings}
places lived: {places_lived}
father's jobs: {fathers_jobs}
mothers's jobs: {mothers_jobs}

#MEMORIES#
{life_memories}

#CHAT HISTORY
{chat_history}
```

Figure 9: Prompt used by the agent being interviewed during a simulated Adult Attachment Interview (AAI) chat.

# E    UMAP PROJECTIONS WITH CLAUDE 3 OPUS

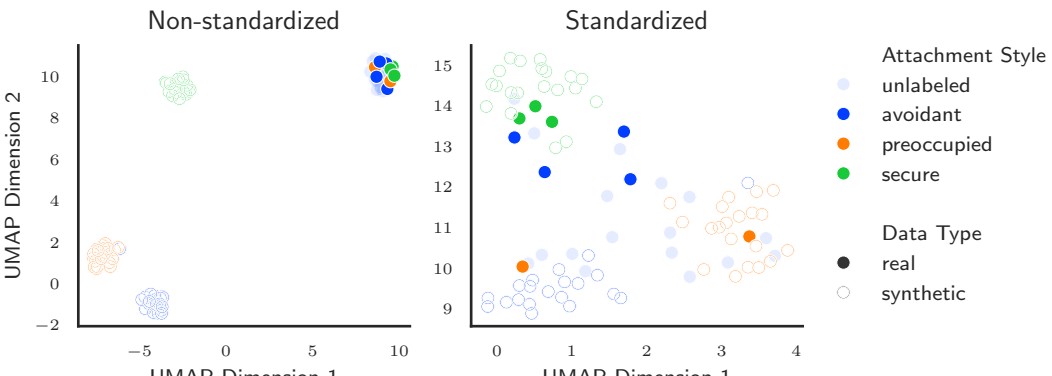

Figure 10: 2D UMAP projections of synthetic (Claude 3 Opus) and human data embeddings. Both plots show clear clusters corresponding to the different attachment styles within the synthetic embeddings. In the left plot, however, the synthetic embeddings occupy a distinct region of the space, separate from the human embeddings. The right plot demonstrates the impact of normalizing synthetic embeddings using unlabeled human data, where the synthetic embeddings are now more closely grouped, and the attachment style clusters better align with the attachment styles of the labeled human dataset interviews.

