# OpenReview forum: "Synthesizing Bonds: Enhancing Adult Attachment Predictions with LLM-Generated Data"
_ICLR.cc/2025/Conference — ICLR 2025 Conference Withdrawn Submission_

### Official Review · Reviewer_RCA7 · 2024-10-25

**Soundness:** 2
**Presentation:** 2
**Contribution:** 2
**Rating:** 5
**Confidence:** 3

**Summary:**

The paper presents a data synthesis approach for predicting attachment styles, powered by a large language model (LLM). It introduces an Interviewee Agent Creation module that generates virtual interviewees with detailed user profiles and childhood memories. This module utilizes Retrieval-Augmented Generation (RAG) to retrieve the most relevant memories, thereby simulating human behavior in the Adult Attachment Interview (AAI).
The authors' key contributions are as follows:
1. They designed a system that simulates human profiles and childhood memories to predict attachment styles.
2. Their data synthesis approach effectively addresses issues of data scarcity and privacy.

**Strengths:**

1. This paper proposed a new framework to apply synthetic data to resolve attachment style prediction issue.
2. The proposed method can resolve the data scarcity and privacy issue effectively.
3. The framework introduced "childhood memories" to increase the diversity of generated synthetic profile.

**Weaknesses:**

1. The value of the proposed task is debatable, and the theory basis of utilizing "childhood memories" for attachment style prediction is not solid enough.
2. The quality of generated "childhood memories", including diversity,  level of detail, objectivity, cannot be guaranteed.

**Questions:**

1. Can you provide more concrete theory basis of why choosing to "childhood memories" as the enrichment resource, but not other features?
2. Are there any other methods to ensure/improve the quality of generated "childhood memories", e.g. improve the diversity, or objectiveness of "childhood memories".

---

### Official Review · Reviewer_PVe3 · 2024-10-30

**Soundness:** 2
**Presentation:** 2
**Contribution:** 2
**Rating:** 3
**Confidence:** 3

**Summary:**

The paper proposed an agent architecture to synthetic user data, which can be used to predict attachment. The method is effective, whose generated data perform comparatively with human data. The experiments prove the effectiveness of this method.

**Strengths:**

1. The paper focuses a significant and interesting topic: medical and metal health.
2. The method is cost-efficient.

**Weaknesses:**

1. The paper proves that LLM generated data can help LLM align with human’s attachment styles. However, it would be more important to explore the ability boundary of those data, since mental health is a significant topic.
2. Your experiments are conducted on two close-sourced models: GPT-4 and Claude 3. Could you please add a experiment on open-source models, such as Llama, Mistral and QwenLM?
3. Regards fair comparison, the baseline model trained on human data is not GPT-based or Claude-based. For comparing the quality of training data, it is important to train on the same base model. Can you train the same base models using human data?

**Questions:**

1. Can you conduct experiments on more datasets?
2. I think more explorations and discussions should be added.

---

### Official Review · Reviewer_7hRq · 2024-11-02

**Soundness:** 2
**Presentation:** 3
**Contribution:** 2
**Rating:** 5
**Confidence:** 3

**Summary:**

This paper explores a novel approach to overcoming the challenge of acquiring real patient data in medical research by using LLMs to generate synthetic data. Authors designed AI agents with distinct profiles and simulated their responses to Adult Attachment Interviews, a tool used to assess how people form emotional connections. They trained predictive models on these synthetic interviews and found that these models could predict attachment styles in real human interviews effectively. Additionally, they improved the alignment between synthetic and real data by using some unlabeled human data. The study demonstrates that synthetic data can be a valuable resource for training models in psychological research, potentially easing the reliance on real patient data.

**Strengths:**

This paper presents a novel application of LLMs to generate synthetic psychological data, specifically targeting the shortage of Adult Attachment Interview (AAI) data. This approach offers a creative solution to a significant issue in clinical psychology.

**Weaknesses:**

1. The validation set of only 9 labeled interviews is extremely small for a 1536-dimensional embedding space. This makes ROC-AUC values potentially unreliable.
2. Statistical significance tests are missing for the performance differences between models, which is essential for validating the claims. Was any statistical power calculation performed to determine if this sample size could detect meaningful effects?
3. The synthetic data generation process lacks quality control metrics. No clear criteria exist for accepting or rejecting generated interviews based on clinical validity.
4. Figure 4 demonstrates that the cosine similarities for GPT-4 generated content are unusually high, ranging from 0.95 to 0.98. This indicates a potential issue of either memorization or a lack of diversity in the synthetic data. This is important because real human interviews usually show more differences. Overall, both models show very high similarities, suggesting that the dataset may require greater diversity in synthetic data generation to more accurately reflect the real-world variation found in human responses.
5. The context window of 4 messages seems arbitrarily small for attachment style analysis. Clinical attachment interviews typically require longer interaction sequences to establish reliable patterns.
6. Could you clarify whether the hyperparameter choices, including the use of 500 estimators in the Extra Trees classifier and temperature settings of 0.7 and 0.5 for sampling, were determined through systematic hyperparameter tuning or ablation studies?
7. The performance metrics for individual attachment styles are not clearly presented. How does the model perform specifically for each attachment style category?
8. The authors only used OpenAI's text-embedding-3-small without comparing different embedding models, which limits our understanding of the method's robustness.

**Questions:**

1. The model's ability to understand attachment patterns relies solely on 9 interviews. How can such a small dataset capture the complexity of attachment theory?
2. The paper shows no evidence that the generated interviews match real clinical patterns. Where is the clinical validation from attachment theory experts?
3. The model might be learning superficial text patterns rather than actual attachment dynamics. What proves the model understands attachment rather than just mimicking language patterns?
4. How do you ensure the model does not generate harmful or clinically inappropriate responses?

---

### Note · Authors · 2024-11-22

I have read and agree with the venue's withdrawal policy on behalf of myself and my co-authors.